# Comparison of the CHU-9D and the EQ-5D-Y instruments in children and young people with cerebral palsy: a cross-sectional study

Jennifer M Ryan ![ORCID],[1,2] Ellen McKay,[2] Nana Anokye,[3] Marika Noorkoiv,[1] Nicola Theis,[4] Grace Lavelle[5]

¹College of Health, Medicine and Life Sciences, Brunel University London, Uxbridge, UK
²Department of Public Health and Epidemiology, Royal College of Surgeons in Ireland, Dublin, Ireland
³Health Economics Research Group, Brunel University London, London, UK
⁴School of Sport and Exercise, University of Gloucestershire, Cheltenham, UK
⁵Institute of Psychiatry, King's College London, London, UK

**Correspondence to**
Dr Jennifer M Ryan;
jennifer.ryan@brunel.ac.uk

## ABSTRACT

**Objective** To compare the performance of the EuroQol 5D youth (EQ-5D-Y) and child health utility 9D (CHU-9D) for assessing health-related quality of life (HRQoL) in children and young people (CYP) with cerebral palsy (CP).

**Design** Cross-sectional study.

**Setting** England.

**Participants** Sixty-four CYP with CP aged 10–19 years in Gross Motor Function Classification System (GMFCS) levels I–III.

**Main outcome measures** Missing data were examined to assess feasibility. Associations between utility values and individual dimensions on each instrument were examined to assess convergent validity. Associations between utility values and GMFCS level were examined to assess known-group differences.

**Results** Missing data were <5% for both instruments. Twenty participants (32.3%) and 11 participants (18.0%) reported full health for the EQ-5D-Y and CHU-9D, respectively. There was poor agreement between utilities from the two instruments (intraclass correlation coefficient=0.62; 95% limits of agreement −0.58 to 0.29). Correlations between EQ-5D-Y and CHU-9D dimensions were weak to moderate (r=0.25 to 0.59). GMFCS level was associated with EQ-5D-Y utility values but not CHU-9D utility values.

**Conclusion** The EQ-5D-Y and CHU-9D are feasible measures of HRQoL in CYP with CP. However, the two instruments demonstrate poor agreement and should not be used to measure and value HRQoL in CYP with CP interchangeably. We propose that the CHU-9D may be preferable to use in this population as it assesses concepts that influence HRQoL among CYP with CP and provides less extreme utility values than the EQ-5D-Y.

## INTRODUCTION

Cerebral palsy (CP) is a heterogeneous disorder characterised by abnormal movement and posture. It is often coexistent with epilepsy, intellectual disability and language, communication or behavioural difficulties.[1] Its prevalence is 2/1000 live births[2]; approximately 110 000 people live with CP in the UK.[3] CP is a lifelong condition. Management of CP

encompasses medical, surgical and rehabilitation interventions.[1] While a large volume of research has examined the effectiveness of interventions for people with CP,[4] there is limited research examining cost-effectiveness of such interventions.

Economic evaluation is used to inform the efficient allocation of resources in a healthcare setting. The cost utility analysis (CUA) is the type of economic evaluation recommended by the National Institute for Health and Care Excellence particularly for interventions funded by National Health Service (NHS) and personal social services.[5] The CUA commonly describes the relationship between costs and health benefits as the cost per quality-adjusted life year (QALY). QALYs incorporate both quantity and quality of life. QALYs are commonly calculated using an assessment of health-related quality of life (HRQoL) obtained from a preference-based instrument. Such instruments can provide a health state utility value, where 0 indicates a health state of equivalent value to being dead and 1.0 indicates full health, by applying a prespecified algorithm based on preferences for health status identified in a specific population.[6]

Generic HRQoL instruments are recommended for use in economic evaluations as they allow comparison across healthcare

interventions and populations. However, if scores from generic measures differ, estimates of cost-effectiveness will be impacted, leading to uncertain conclusions regarding whether an intervention provides value for money. A number of generic measures have been used to obtain health state utility values from paediatric populations.[7] The most commonly used are the EuroQol 5D (EQ-5D), the child health utility 9D (CHU-9D), the EQ-5D youth (EQ-5D-Y) and the Health Utilities Index 2 and 3 (HUI-2 and HUI-3).[7] To date, however, only the HUI-2 and HUI-3 have been used to elicit utility values from children and young people (CYP) with CP and significant variation was reported in these.[8]

The EQ-5D-Y and CHU-9D are two generic preference-based HRQoL instruments that were designed specifically for young people. The EQ-5D-Y is a youth-modified version of the adult instrument, the EQ-5D, which was developed by revising the content and wording of the adult instrument.[9] The CHU-9D was developed from the outset for young people based on in-depth interviews with young people with chronic and acute health conditions.[10] Although the performance of the EQ-5D-Y and CHU-9D has been examined in an adolescent population,[11] their performance has not been examined among CYP with CP.

Given the financial cost of CP per annum is approximately AU\$1.5 billion[12] and a wide range of interventions are currently available for people with CP,[4] there is an increasing need for economic evaluation in this area. Prior to conducting economic evaluations, the performance of the EQ-5D-Y and CHU-9D in CYP with CP requires evaluation. The aim of this study was to compare the performance of the EQ-5D-Y and CHU-9D for assessing HRQoL in a community-based sample of CYP with CP. Specific objectives were to examine the feasibility of administering the instruments, to examine convergent validity and to examine known-group differences for both instruments.

## METHODS
### Sample
CYP with CP who participated in a randomised controlled trial examining the effects of progressive resistance training were included in this study.[13] Data collected at baseline were used for this cross-sectional study. Participants were recruited from eight NHS trusts in England, a special education needs school, a university and a primary care organisation in London, national organisations for people with disabilities, and by word of mouth. Inclusion criteria for participation in the trial were: aged 10–19 years with spastic CP and the ability to walk independently with or without a mobility aid (ie, Gross Motor Function Classification System (GMFCS) levels I–III). Exclusion criteria for participation in the trial were orthopaedic surgery of the lower limbs in the past 12 months, botulinum toxin type A injections or serial casting in the past 6 months and insufficient cognition to comply with assessment procedures and the training programme. Participants 16 years and older provided written consent.

Those under 16 years provided assent alongside written consent from a parent or guardian.

Data on the person's demographics, condition and HRQoL were collected using standardised questionnaires during an interview with a researcher at one time point. Both HRQoL questionnaires were self-administered to all participants using the standardised instructions accompanying each instrument. Assistance was provided by the researcher to read the questions if required. Further, the young person was allowed to ask their parent/guardian or researcher for assistance to answer the questions if required. Anatomical distribution was described as unilateral or bilateral.[14] Functional mobility was classified according to the GMFCS. The GMFCS is a five-level classification system, where level I indicates most able and level V indicates most limited. Those in GMFCS level I are able to walk and run and climb stairs without assistance. Those in level II are able to walk in most settings but may use a hand-held mobility device indoors or wheeled mobility to travel long distances. Those in level III can walk using a hand-held mobility device but use a wheelchair or powered mobility outdoors. Participants selected a statement that best described their mobility based on descriptors of each GMFCS level.[15] Two physiotherapists retrospectively cross-referenced subjective ratings of GMFCS level against video recordings of participants, obtained as part of the baseline assessment.

### Utility measurement
The EQ-5D-Y assesses a person's health across five dimensions. The five dimensions are 'mobility', 'looking after myself', 'doing usual activities', 'having pain or discomfort' and 'feeling worried, sad or unhappy'.[9] Each dimension is rated on one of three levels (no problems, some problems and a lot of problems) that describes a person's health today.[9] The EQ-5D-Y was developed by reviewing the applicability of the EQ-5D domain concepts and wording for children and adolescents.[9] The EQ-5D-Y is suitable for use in young people aged 8–19 years.[16] At present, there is no value set for the EQ-5D-Y. Although use of the adult value set (EQ-5D) for the EQ-5D-Y is not recommended, we calculated utilities using UK-based adult weights.[17] We acknowledge the limitation of this, but considered it to be the best method given the lack of weights for the EQ-5D-Y. Further, although the performance of the EQ-5D-Y using the adult value set has been examined in CYP with typical development,[18] it has not been examined in a clinical population.

There are nine dimensions in the CHU-9D: 'worried', 'sad', 'pain', 'tired', 'annoyed', 'schoolwork', 'sleep', 'daily routine' and 'ability to join in activities'.[10] For each dimension, the person describes how they are today according to one of five levels based on severity (eg, not worried, a little bit worried, a bit worried, quite worried, very worried), which were determined from in-depth interviews with young people.[10] The CHU-9D is suitable for use in young people aged 7–17 years.[10 19] The scoring algorithm based on the preferences of the UK

adult general population was used to estimate utilities.[20] This method has been shown to be appropriate to use for young people.[18]

## Patient and public involvement

CYP with and without CP were involved in the design, conduct and dissemination plans of our research relating to the randomised controlled trial examining the effects of progressive resistance training.

## Statistical analysis

The distribution of data was examined using histograms, Q–Q plots and cross-tabulations. Mean and SD, median and IQR, frequencies and percentages were used to report the data as appropriate. We examined feasibility by reporting the number of participants with missing data and the percentage of missing data for each instrument. The instrument was considered feasible if missing data were <5%.[21] Participants who were missing data for an individual dimension of the EQ-5D-Y or CHU-9D were excluded from the calculation of utility values and from analyses involving that dimension.

To assess convergent validity, we calculated an intra-class correlation coefficient (ICC) between CHU-9D and EQ-5D-Y utility values. We interpreted an ICC >0.75 as indicating good agreement.[22] We compared mean utility between instruments using linear regression with a bootstrap procedure as there was evidence that residuals were not normally distributed. Bias corrected and accelerated bootstrap CIs were calculated from 2000 replicates.[23] We also produced a Bland-Altman plot of the difference between the two instruments against their mean to examine agreement between utilities from the two instruments. We calculated 95% limits of agreement as mean difference ±1.96 SD.[24] We examined the association between (1) CHU-9D utility value and levels of each EQ-5D-Y dimension and (2) EQ-5D-Y utility value and levels of each CHU-9D dimension, by calculating Spearman's correlation coefficients. We also examined associations between each dimension of the EQ-5D-Y and CHU-9D by calculating Spearman's correlation coefficients. Based on the description of each dimension and associations observed among adolescents with typical development,[11] we hypothesised that the following dimensions would be correlated between the CHU-9D and EQ-5D-Y, respectively: 'worried' versus 'feeling worried, sad or unhappy', 'sad' versus 'feeling worried, sad or unhappy', 'pain' versus 'having pain/discomfort', 'daily routine' versus 'doing usual activities', 'daily routine' versus 'looking after myself', 'able to join in' versus 'doing usual activities' and 'able to join in' versus 'mobility'. To aid interpretation, we proposed a correlation of 0.10–0.39 to indicate a weak association, a correlation of 0.40–0.75 to indicate a moderate association and a correlation of >0.75 to indicate a strong association.[22 25] However, this interpretation should be used with caution given that cut-offs for interpreting correlation coefficients are arbitrary and may be inconsistent with other studies.[25]

We examined known-group differences by fitting linear regression models using a bootstrap procedure to compare CHU-9D utility values and EQ-5D-Y utility values, respectively, across functional mobility as defined by the GMFCS. A bootstrap procedure was used as there was evidence that residuals were not normally distributed. For each model, utility value was the dependent variable and bias corrected and accelerated bootstrap CIs were calculated from 2000 replicates.[23] We also compared the number of people experiencing no problems versus any problems for each CHU-9D and EQ-5D-Y dimension across GMFCS level, using a $\chi^2$ test. It was expected that CYP with better functional mobility would have higher utilities.[26 27] MedCalc V.19.2.0 was used to produce the Bland-Altman plot. All other statistical analyses were performed using Stata V.13.

## RESULTS

Sixty-four participants were recruited to the study. One person did not complete the EQ-5D-Y or CHU-9D. Therefore, 63 participants were included in the analysis. Table 1 describes the participant characteristics. The mean±SD age was 13.7±2.5 years. The majority of participants (86%) was in GMFCS levels I or II indicating a mild lower limb impairment. The majority of participants was White British (59%) and attended a mainstream school (71%).

For the EQ-5D-Y, one participant did not provide a response to the 'looking after myself' dimension. For the CHU-9D, two participants did not provide a response to the schoolwork dimension and one participant did not provide a response to the ability to 'join in activities' dimension. Utility values for the EQ-5D-Y were therefore calculated for 62 out of 64 participants (96.9%) and utility values for the CHU-9D were calculated for 61 out of 64 participants (95.3%). Missing data were 1.9% for the EQ-5D-Y and 2.0% for the CHU-9D.

The distribution of EQ-5D-Y and CHU-9D utilities is shown in figures 1 and 2. For the EQ-5D-Y and CHU-9D, respectively, 20 participants (32.3%) and 11 participants (18.0%) reported full health. The median (IQR) EQ-5D-Y utility value was 0.80 (0.62–1.00). The mean±SD EQ-5D-Y utility value was 0.73±0.29 (range −0.17 to 1.00). Two participants (3.2%) reported a 'worse than death' health state for the EQ-5D-Y. The median (IQR) CHU-9D utility value was 0.92 (0.83–0.96). The mean±SD CHU-9D utility value was 0.89±0.10 (range 0.56–1.00). There was poor agreement between utilities from the two instruments as indicated by an ICC of 0.62. The CHU-9D utility value was on average 0.15 (95% CI 0.09 to 0.25) higher than the EQ-5D-Y utility value. 95% limits of agreement were −0.58 to 0.29, indicating that the EQ-5D-Y utility value may be 0.29 higher or 0.58 lower than the CHU-9D utility value (figure 3).

The percentages of reported problems across dimensions of the EQ-5D-Y and CHU-9D are presented in tables 2 and 3. There was evidence of correlation between

**Table 1** Participant characteristics

| | n (%) | Mean (SD) | Range | Mean (SD) EQ-5D-Y utility value | Median (IQR) EQ-5D-Y utility value | Mean (SD) CHU-9D utility value | Median (IQR) CHU-9D utility value |
|---|---|---|---|---|---|---|---|
| Age, years | 63 | 13.7 (2.5) | 10–19 | – | – | – | – |
| Female | 26 (41.3) | | | 0.77 (0.29) | 0.83 (0.69–1.00) | 0.90 (0.08) | 0.92 (0.83–0.97) |
| Male | 37 (58.7) | | | 0.71 (0.29) | 0.77 (0.62–0.93) | 0.88 (0.11) | 0.92 (0.81–0.95) |
| Height, cm | 63 | 154.3 (12.7) | 131.5–180.9 | – | – | – | – |
| Mass, kg | 63 | 49.3 (13.7) | 27.4–78.5 | – | – | – | – |
| Ethnicity | | | | | | | |
| White British | 37 (58.7) | | | 0.80 (0.20) | 0.81 (0.69–1.00) | 0.89 (0.09) | 0.92 (0.84–0.95) |
| Black or Black British | 5 (7.9) | | | 0.62 (0.44) | 0.81 (0.19–1.00) | 0.90 (0.10) | 0.90 (0.81–1.00) |
| Asian or Asian British | 9 (14.3) | | | 0.59 (0.49) | 0.73 (0.20–1.00) | 0.88 (0.12) | 0.92 (0.89–0.95) |
| Other | 12 (19.0) | | | 0.69 (0.22) | 0.75 (0.57–0.81) | 0.87 (0.13) | 0.87 (0.82–0.97) |
| GMFCS level | | | | | | | |
| I | 29 (46.0) | | | 0.82 (0.20) | 0.81 (0.73–1.00) | 0.89 (0.11) | 0.92 (0.82–0.95) |
| II | 25 (39.7) | | | 0.75 (0.25) | 0.75 (0.62–1.00) | 0.90 (0.08) | 0.92 (0.86–0.97) |
| III | 9 (14.3) | | | 0.39 (0.43) | 0.44 (0.00–0.76) | 0.82 (0.12) | 0.83 (0.72–0.89) |
| Distribution | | | | | | | |
| Unilateral | 31 (49.2) | | | 0.81 (0.18) | 0.80 (0.69–1.00) | 0.91 (0.09) | 0.92 (0.87–0.98) |
| Bilateral | 32 (50.8) | | | 0.66 (0.35) | 0.75 (0.59–1.00) | 0.87 (0.11) | 0.87 (0.81–0.95) |
| Type of school | | | | | | | |
| Mainstream school | 45 (71.4) | | | 0.77 (0.24) | 0.80 (0.67–1.00) | 0.88 (0.08) | 0.90 (0.82–0.95) |
| SEN | 13 (20.6) | | | 0.62 (0.41) | 0.80 (0.44–0.85) | 0.89 (0.11) | 0.92 (0.86–0.98) |
| Further college education | 2 (3.2) | | | 0.85 (0.22) | 0.85 (0.69–1.00) | 0.99 (0.01) | 0.99 (0.98–1.00) |
| University | 3 (4.8) | | | 0.67 (0.45) | 0.85 (0.16–1.00) | 0.85 (0.25) | 1.00 (0.56–1.00) |

CHU-9D, child health utility 9D; EQ-5D-Y, EuroQol 5D youth; GMFCS, gross motor function classification system; SEN, special education needs.

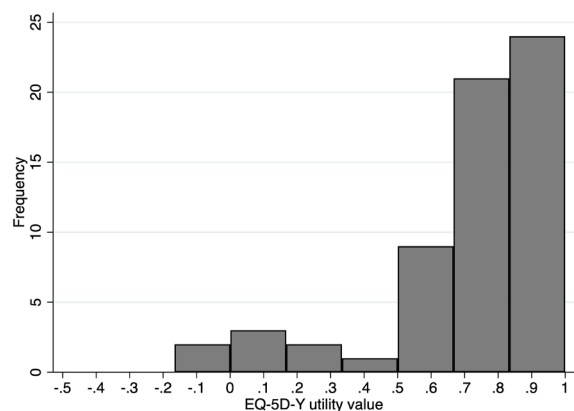

**Figure 1** Distribution of utility values for the EuroQol 5D youth (EQ-5D-Y).

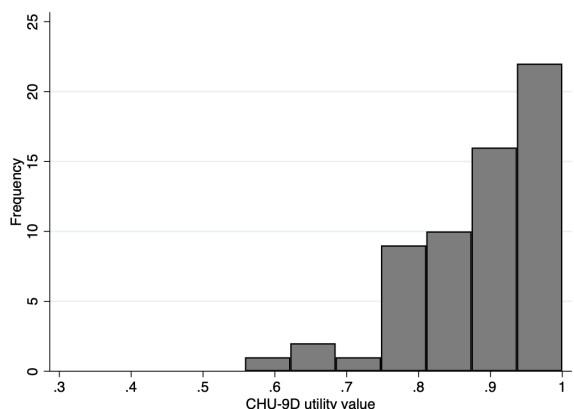

**Figure 2** Distribution of utility values for the child health utility 9D (CHU-9D).

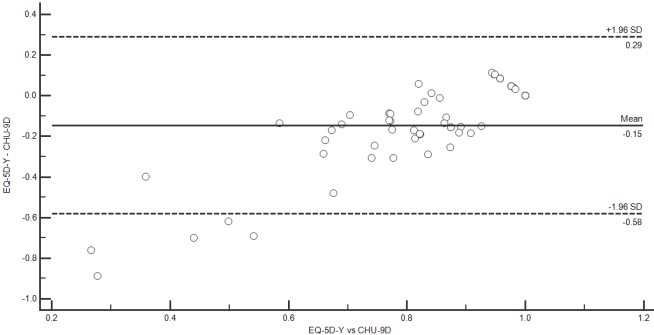

**Figure 3** Bland-Altman plot of differences between EQ-5D-Y and CHU-9D utility values against average of EQ-5D-Y and CHU-9D utility values. CHU-9D, child health utility 9D; EQ-5D-Y, EuroQol 5D youth.

EQ-5D-Y utility values and level of severity for all dimensions of the CHU-9D except for the 'sad' dimension. Correlations ranged from −0.12 to −0.64 (table 2). For all dimensions of the EQ-5D-Y, the median CHU-9D utility value decreased with increasing levels of severity on the EQ-5D-Y dimensions (table 3). For all dimensions of the EQ-5D-Y, the majority of respondents reported themselves in the least severe level (ie, no problems). However, responses for most dimensions of the EQ-5D-Y were spread across all levels, from no problems to a lot of problems. The exception to this was the 'worried, sad or unhappy' dimension; no participant reported feeling very worried, sad or unhappy. Similarly, for all dimensions of the CHU-9D except for 'tired', the majority of participants reported themselves in the least severe level. However, responses were not spread across all levels for each dimension. For the 'worried', 'sad', 'pain', 'annoyed', 'sleep' and 'daily routine' dimensions of the CHU-9D, no participant reported the most severe level.

Correlations between CHU-9D and EQ-5D-Y dimensions are presented in table 4. Moderate correlations (r=0.43–0.59) were observed for all hypothesised associations, except for 'worried' versus 'feeling worried, sad or unhappy' (r=0.25) and 'daily routine' versus 'doing usual activities' (r=0.35). Several unexpected correlations were observed between dimensions. Namely, 'schoolwork' on the CHU-9D was associated with 'mobility' (r=0.56), 'pain or discomfort' (r=0.55) and 'doing usual activities' (r=0.44) on the EQ-5D-Y; 'tired' on the CHU-9D was associated with 'pain/discomfort' on the EQ-5D-Y (r=0.40) and 'annoyed' on the CHU-9D was associated with 'worried/sad/unhappy' on the EQ-5D-Y (r=0.53).

Median (IQR) utilities by GMFCS level are presented in table 1. EQ-5D-Y utility value was associated with GMFCS level ($R^2$=0.231, p=0.016; online supplemental table 1). As expected, EQ-5D-Y utility value was on average 0.43 lower in individuals in level III compared with those in level I (95% CI 0.14 to 0.73) and 0.36 lower in individuals in level III compared with those in level II (95% CI 0.06 to 0.66). However, there was no difference in utility value between those in levels I and II. Although median

CHU-9D utility value was lower in GMFCS level III compared with GMFCS levels I and II (table 1), there was no evidence that CHU-9D utility value was associated with GMFCS level ($R^2$=0.071, p=0.170; online supplemental table 2). As presented in table 5, the percentage of people reporting some or a lot of problems on the 'mobility' and 'looking after myself' dimensions of the EQ-5D-Y differed according to GMFCS level, with the percentage of people reporting problems increasing from GMFCS levels I to III. The percentage of people reporting problems was not associated with GMFCS level for any other EQ-5D-Y dimension. There was also no evidence of an association between the percentage of people reporting problems and GMFCS level for any dimension of the CHU-9D (table 6).

## DISCUSSION

The aim of this study was to compare the performance of the EQ-5D-Y and CHU-9D for assessing HRQoL in CYP with CP. Specific objectives were to examine feasibility, convergent validity and known-group differences. To our knowledge, this is the first study to compare the EQ-5D-Y and CHU-9D in a clinical population. Although the results indicate that the two instruments are feasible to use when administered by interview to CYP with CP, the two instruments have poor agreement and may not be used interchangeably to measure and value HRQoL among CYP with CP. The EQ-5D-Y utility value was on average 0.15 lower than the CHU-9D utility value. However, there was considerable variation in individual differences between instruments. 95% limits of agreement demonstrated that the EQ-5D-Y utility value may be 0.29 higher or 0.58 lower than the CHU-9D utility value for an individual. Additionally, dimensions on each instrument that were hypothesised to measure similar concepts were only weakly to moderately associated.

Two studies compared the EQ-5D-Y and CHU-9D in CYP with typical development.[11 18] Agreement between EQ-5D-Y and CHU-9D utility values was better among adolescents with typical development than among CYP with CP, with an ICC of 0.80 and much narrower 95% limits of agreement (−0.268 to 0.241) reported.[11] Although the median EQ-5D-Y utility value among CYP in this study was similar to that reported for Australian adolescents with typical development (0.80 vs 0.83), the median CHU-9D utility was higher (0.92 vs 0.83).[11] The median CHU-9D utility reported by CYP with CP was more similar to that reported by children, aged 6–7 years, with typical development living in England (0.92 vs 0.90) and the median EQ-5D-Y utility was identical between these groups.[18] This may be because the study of children living in England used the same algorithms to estimate utilities as the current study, while the study of adolescents used algorithms developed in the Australian population. It is also plausible that CYP with CP have better HRQoL than CYP with typical development. A study of a large sample of CYP with CP across Europe found that CYP with CP

**Table 2**  Number and percentage of participants reporting problems across CHU-9D dimensions

| CHU-9D dimensions | Levels | Frequency (%) | n | Median±IQR EQ-5D-Y utility value | Rho | P value |
|---|---|---|---|---|---|---|
| Worried | Not worried | 82.5 | 52 | 0.81±0.29 | −0.38 | 0.002 |
| | A little bit | 15.9 | 10 | 0.55±0.66 | | |
| | A bit | 1.6 | 1 | −0.11 | | |
| | Quite | 0 | 0 | – | | |
| | Very | 0 | 0 | – | | |
| Sad | Not sad | 93.7 | 59 | 0.80±0.34 | −0.12 | 0.350 |
| | A little bit | 3.2 | 2 | 0.50±0.69 | | |
| | A bit | 3.2 | 2 | 0.68±0.33 | | |
| | Quite | 0 | 0 | – | | |
| | Very | 0 | 0 | – | | |
| Pain | No pain | 61.9 | 39 | 0.85±0.29 | −0.42 | <0.001 |
| | A little bit | 30.2 | 19 | 0.73±0.23 | | |
| | A bit | 3.2 | 2 | 0.39±0.60 | | |
| | Quite | 4.8 | 3 | 0.16±0.73 | | |
| | A lot | 0 | 0 | – | | |
| Tired | Not tired | 36.5 | 23 | 0.82±0.27 | −0.44 | <0.001 |
| | A little bit | 42.9 | 27 | 0.81±0.31 | | |
| | A bit | 9.5 | 6 | 0.59±0.07 | | |
| | Quite | 7.9 | 5 | 0.19±0.46 | | |
| | Very | 3.2 | 2 | 0.45±0.52 | | |
| Annoyed | Not annoyed | 84.1 | 53 | 0.81±0.30 | −0.27 | 0.034 |
| | A little bit | 11.1 | 7 | 0.62±0.33 | | |
| | A bit | 3.2 | 2 | 0.50±0.69 | | |
| | Quite | 1.6 | 1 | 0.52 | | |
| | Very | 0 | 0 | – | | |
| Schoolwork/homework | No problems | 63.5 | 40 | 0.93±0.21 | −0.64 | <0.001 |
| | Few problems | 17.5 | 11 | 0.68±0.10 | | |
| | Some problems | 14.3 | 9 | 0.55±0.52 | | |
| | Many problems | 0 | 0 | – | | |
| | Can not do schoolwork | 1.6 | 1 | 0.20 | | |
| | Missing | 3.2 | 2 | – | | |
| Sleep | No problems | 71.4 | 45 | 0.81±0.29 | −0.38 | 0.002 |
| | Few problems | 20.6 | 13 | 0.70±0.24 | | |
| | Some problems | 4.8 | 3 | 0.73±0.07 | | |
| | Many problems | 3.2 | 2 | 0.02±0.27 | | |
| | Can not sleep | 0 | 0 | – | | |
| Daily routine | No problems | 74.6 | 47 | 0.81±0.27 | −0.48 | <0.001 |
| | Few problems | 15.9 | 10 | 0.52±0.59 | | |
| | Some problems | 7.9 | 5 | 0.66±0.14 | | |
| | Many problems | 1.6 | 1 | 0.09 | | |
| | Can not do daily routine | 0 | 0 | – | | |

Continued

**Table 2** Continued

| CHU-9D dimensions | Levels | Frequency (%) | n | Median±IQR EQ-5D-Y utility value | Rho | P value |
|---|---|---|---|---|---|---|
| Able to join in activities | Any activities | 57.1 | 36 | 0.85±0.27 | −0.43 | <0.001 |
| | Most activities | 27.0 | 17 | 0.73±0.23 | | |
| | Some activities | 7.9 | 5 | 0.60±0.33 | | |
| | Few activities | 4.8 | 3 | 0.71±0.33 | | |
| | No activities | 1.6 | 1 | −0.11 | | |
| | Missing | 1.6 | 1 | – | | |

CHU-9D, child health utility 9D; EQ-5D-Y, EuroQol 5D youth.

had better QoL in five domains (moods and emotions, self-perception, autonomy, relationships with parents and school life) compared with CYP in the general population matched for age, sex and country.[28]

Although the EQ-5D-Y utility value was on average lower than the CHU-9D utility value, 32% of participants reported full health when using the EQ-5D-Y compared with only 18% of participants when using the CHU-9D. A similar ceiling effect for the EQ-5D-Y was reported among CYP with typical development.[11] Although a large proportion of CYP had 'perfect health' according to the EQ-5D-Y, two children reported a health state worse than death. Neither of these children reported the worst possible health state according to the CHU-9D. The CHU-9D may be more sensitive to varying severities of health because of the larger number of levels for each dimension. The CHU-9D is also the only generic measure that was specifically designed for CYP and the levels were determined

from in-depth interviews with young people.[10] Further, as suggested previously, the extreme health states observed for the EQ-5D-Y may be a result of misapplication of the adult EQ-5D tariff to the health states defined by the EQ-5D-Y.[18] The findings from this study provide additional evidence that the EQ-5D-Y should not be used to measure utilities until a specific value set for the EQ-5D-Y is available.

Although we found associations between a number of dimensions on the EQ-5D-Y and CHU-9D as hypothesised, these associations were only weak to moderate, despite both instruments being administered at the same time point, by the same researcher and in the same environment. These correlations were weaker than those reported among Australian adolescents with typical development.[11] In particular, the correlation between 'feeling worried, sad or unhappy' on the EQ-5D-Y and 'worried' on the CHU-9D was 0.70 among adolescents with typical

**Table 3** Number and percentage of participants reporting problems across EQ-5D-Y dimensions

| EQ-5D-Y dimensions | Levels | Frequency (%) | n | Median ±IQR CHU-9D utility value | Rho | P value |
|---|---|---|---|---|---|---|
| Mobility | No | 50.8 | 32 | 0.95±0.09 | −0.52 | <0.001 |
| | Some | 44.4 | 28 | 0.84±0.13 | | |
| | A lot of | 4.8 | 3 | 0.75±0.05 | | |
| Looking after myself | No | 66.7 | 42 | 0.94±0.09 | −0.46 | <0.001 |
| | Some | 27.0 | 17 | 0.83±0.10 | | |
| | A lot of | 4.8 | 3 | 0.72±0.27 | | |
| | Missing | 1.6 | 1 | 0.78 | | |
| Usual activities | No | 66.7 | 42 | 0.93±0.11 | −0.54 | <0.001 |
| | Some | 28.6 | 18 | 0.83±0.15 | | |
| | A lot of | 4.8 | 3 | 0.72±0.16 | | |
| Pain or discomfort | No | 55.6 | 35 | 0.95±0.11 | −0.54 | <0.001 |
| | Some | 38.1 | 24 | 0.86±0.16 | | |
| | A lot of | 6.3 | 4 | 0.72±0.24 | | |
| Worried, sad or unhappy | Not | 84.1 | 53 | 0.92±0.11 | −0.44 | <0.001 |
| | A bit | 15.9 | 10 | 0.79±0.05 | | |
| | Very | 0 | 0 | – | | |

CHU-9D, child health utility 9D; EQ-5D-Y, EuroQol 5D youth.

**Table 4** Spearman rank correlation coefficients between CHU-9D and EQ-5D-Y dimensions

| EQ-5D-Y dimensions | | Worried | Sad | Pain | Tired | Annoyed | Schoolwork | Sleep | Daily routine | Activities |
|---|---|---|---|---|---|---|---|---|---|---|
| Mobility | r | **0.30** | −0.01 | **0.35** | **0.36** | 0.19 | **0.56*** | **0.31** | 0.19 | **0.49†** |
| | p | **0.017** | 0.950 | **0.006** | **0.004** | 0.140 | **<0.001** | **0.013** | 0.144 | **<0.001** |
| Looking after myself | r | **0.26†** | −0.04† | 0.21† | 0.22† | 0.08† | **0.39‡** | **0.27†** | **0.59†** | **0.36*** |
| | p | **0.040** | 0.741 | 0.098 | 0.086 | 0.525 | **0.002** | **0.031** | **<0.001** | **0.004** |
| Activities | r | **0.36** | 0.08 | **0.30** | 0.23 | **0.25** | **0.44*** | **0.38** | **0.35** | **0.54†** |
| | p | **0.004** | 0.543 | **0.017** | 0.065 | **0.049** | **<0.001** | **0.002** | **0.006** | **<0.001** |
| Pain or discomfort | r | **0.33** | 0.07 | **0.47** | **0.40** | 0.22 | **0.55*** | **0.39** | **0.25** | **0.32†** |
| | p | **0.008** | 0.600 | **<0.001** | **0.001** | 0.090 | **<0.001** | **0.002** | **0.048** | **0.011** |
| Worried, sad, unhappy | r | **0.25** | **0.43** | 0.19 | **0.26** | **0.53** | **0.39*** | −0.01 | **0.33** | 0.15† |
| | p | **0.048** | **<0.001** | 0.143 | **0.039** | **<0.001** | **0.002** | 0.944 | **0.008** | 0.239 |

Correlations with p value<0.05 highlighted in bold; n=63 for all analyses unless stated otherwise.
*n=61.
†n=62.
‡n=60.
CHU-9D, child health utility 9D; EQ-5D-Y, EuroQol 5D youth.

development, compared to 0.25 among CYP with CP. This suggests that CYP with CP interpret these two dimensions differently and they do not measure the same concept in this population. The strongest correlation between EQ-5D-Y and CHU-9D dimensions among adolescents with typical development was between 'having pain or discomfort' and 'pain' (r=0.753).[11] However, 'having pain or discomfort' and 'pain' were only moderately correlated in CYP with CP (r=0.47). Many CYP with CP

experience frequent pain,[29] and as a result, their interpretation of the EQ-5D-Y question about pain and discomfort may differ to their interpretation of the CHU-9D question about pain only. Indeed, 56% of CYP with CP reported no pain or discomfort on the EQ-5D-Y, while 62% reported no pain on the CHU-9D. Of the hypothesised correlations, only the correlation between 'looking after myself' on the EQ-5D-Y and 'daily routine' on the CHU-9D was stronger among CYP with CP compared with adolescents

**Table 5** Percentage reporting problems on each EQ-5D-Y dimension across GMFCS level

| EQ-5D-Y dimensions | Levels | GMFCS level I n=29 | GMFCS level II n=25 | GMFCS level III n=9 | P value* |
|---|---|---|---|---|---|
| Mobility | No | 69% | 44% | 11% | 0.007 |
| | Some | 31% | 56% | 56% | |
| | A lot of | 0% | 0% | 33% | |
| Looking after myself | No | 83% | 56% | 50% | 0.036 |
| | Some | 17% | 40% | 25% | |
| | A lot of | 0% | 4% | 25% | |
| Usual activities | No | 72% | 72% | 33% | 0.072 |
| | Some | 28% | 24% | 44% | |
| | A lot of | 0% | 4% | 22% | |
| Pain or discomfort | No | 59% | 60% | 33% | 0.348 |
| | Some | 38% | 36% | 44% | |
| | A lot of | 3% | 4% | 22% | |
| Worried, sad or unhappy | No | 79% | 92% | 78% | 0.380 |
| | Some | 21% | 8% | 22% | |
| | A lot of | 0% | 0% | 0% | |

*Comparing number of people experiencing 'no' problems versus any problems (ie, some and a lot of) across GMFCS levels.
EQ-5D-Y, EuroQol 5D youth; GMFCS, Gross Motor Function Classification System.

**Table 6** Percentage reporting problems on each CHU-9D dimension across GMFCS level

| CHU-9D dimensions | Levels | GMFCS level I<br>n=29 | GMFCS level II<br>n=25 | GMFCS level III<br>n=9 | P value* |
|---|---|---|---|---|---|
| Worried | Not worried | 83% | 84% | 78% | 0.914 |
| | A little bit | 17% | 16% | 11% | |
| | A bit | 0% | 0% | 11% | |
| | Quite | 0% | 0% | 0% | |
| | Very | 0% | 0% | 0% | |
| Sad | Not sad | 86% | 100% | 100% | 0.082 |
| | A little bit | 7% | 0% | 0% | |
| | A bit | 7% | 0% | 0% | |
| | Quite | 0% | 0% | 0% | |
| | Very | 0% | 0% | 0% | |
| Pain | No pain | 55% | 72% | 56% | 0.408 |
| | A little bit | 38% | 24% | 22% | |
| | A bit | 0% | 4% | 11% | |
| | Quite | 7% | 0% | 11% | |
| | A lot | 0% | 0% | 0% | |
| Tired | Not tired | 31% | 44% | 33% | 0.601 |
| | A little bit | 52% | 32% | 44% | |
| | A bit | 7% | 12% | 11% | |
| | Quite | 10% | 8% | 0% | |
| | Very | 0% | 4% | 11% | |
| Annoyed | Not annoyed | 72% | 96% | 89% | 0.056 |
| | A little bit | 17% | 4% | 11% | |
| | A bit | 7% | 0% | 0% | |
| | Quite | 4% | 0% | 0% | |
| | Very | 0% | 0% | 0% | |
| Schoolwork/homework | No problems | 72% | 64% | 43% | 0.104 |
| | Few problems | 17% | 16% | 29% | |
| | Some problems | 10% | 20% | 14% | |
| | Many problems | 0% | 0% | 14% | |
| | Can not do schoolwork | 0% | 0% | 0% | |
| Sleep | Not | 79% | 68% | 56% | 0.343 |
| | A little bit | 10% | 28% | 33% | |
| | A bit | 7% | 4% | 0% | |
| | Quite | 4% | 0% | 11% | |
| | Very | 0% | 0% | 0% | |
| Daily routine | Not | 83% | 72% | 56% | 0.243 |
| | A little bit | 10% | 20% | 22% | |
| | A bit | 7% | 4% | 22% | |
| | Quite | 0% | 4% | 0% | |
| | Very | 0% | 0% | 0% | |
| Able to join in activities | Not | 69% | 52% | 38% | 0.135 |
| | A little bit | 24% | 28% | 38% | |
| | A bit | 3.5% | 12% | 13% | |
| | Quite | 3.5% | 8% | 0% | |
| | Very | 0% | 0% | 13% | |

*Comparing number of people experiencing 'no' problems versus any problems (ie, a little bit, a bit, quite and very) across GMFCS levels.
CHU-9D, child health utility 9D; GMFCS, Gross Motor Function Classification System.

with typical development.[11] As the 'looking after myself' dimension refers specifically to washing and dressing, this suggests that CYP with CP interpret self-care as part of their daily routine.

We also observed a number of unexpected associations between EQ-5D-Y and CHU-9D dimensions. 'School-work' on the CHU-9D was associated with 'mobility', 'having pain or discomfort' and 'doing usual activities' on the EQ-5D-Y. Approximately 45% of CYP with CP have an intellectual disability, which may range from mild to severe.[30] Although we excluded individuals with insufficient cognition to comply with assessment procedures, some participants may have had a mild intellectual disability. As CYP with intellectual disability are likely to have more severely impaired physical functioning,[30] it is possible that CYP with intellectual disability have more problems completing schoolwork and more problems with mobility and doing usual activities. Alternatively, CYP without intellectual disability but with severe physical impairment, who will have more problems with mobility and doing usual activities, may also have more problems with completing schoolwork as a result of their physical impairment.

We observed an association between EQ-5D-Y utility values and GMFCS level but not between CHU-9D utility values and GMFCS level. Two studies reported that utility values obtained from the HUI-3 differed according to GMFCS level.[26 27] However, regardless of the instrument used, we found that those in GMFCS levels I and II have a similar mean utility value, while those in GMFCS level III have a lower mean utility value compared with levels I and II. The lack of statistical evidence of an association between CHU-9D utility value and GMFCS level may be due to the narrower utility range of the CHU-9D, which resulted in smaller incremental changes between levels. There was also a small number of participants in GMFCS level III, which likely resulted in reduced statistical power to detect differences between groups. However, it is also plausible that HRQoL is not associated with functional mobility. When condition-specific measures of QoL were used, associations between QoL and GMFCS level were not observed.[27 28]

The mean EQ-5D-Y utility value for each GMFCS level was similar to values obtained from the HUI-3 in one study (0.84, 0.50 and 0.39 for adolescents in GMFCS levels I, II and III, respectively)[27] but not similar to those in a second study (0.67, 0.59 and 0.43 in GMFCS levels I, II and III, respectively).[26] The mean CHU-9D utility values for each GMFCS level were higher than those obtained from the HUI-3.[26 27] Discrepancies may be due to differences in the algorithms used to derive utility values. HUI-3 utility values were derived in one study using preferences from the general adult population in Canada.[27] The second study did not state the preferences used to derive HUI-3 utility values.[26] Differences in utilities between the CHU-9D and HUI-3 do not necessarily indicate that the CHU-9D is inaccurate. The HUI-3 was only weakly correlated with a condition-specific measure of HRQoL

in CYP with CP.[27] This weak correlation highlights that condition-specific measures and generic measures of HRQoL may not assess similar concepts. However, unlike generic measures, condition-specific measures are not recommended for evaluations across different conditions.[31] Although a generic measure, the CHU-9D may be more likely than the EQ-5D-Y and HUI to capture dimensions of HRQoL that are important to CYP with CP. In particular, the CHU-9D includes more questions about psychological difficulties and pain, which are associated with QoL among CYP with CP.[28]

## Limitations

The use of adult weights to value EQ-5D-Y health states is a limitation of this study. It has previously been demonstrated that values for health states ascribed by adults differ to corresponding health states ascribed by children.[32 33] However, we used adult weights in the absence of value sets for the EQ-5D-Y. The findings of this study are limited by a small sample. In particular, there were a small number of participants in GMFCS level III. Although the sample included CYP with CP in GMFCS levels I-III, which represents about 70% of individuals with CP,[34] the findings may not be applicable to non-ambulatory CYP. As the sample volunteered to participate, they may have a higher HRQoL than those who did not volunteer.

In conclusion, the findings of this study illustrate that the EQ-5D-Y and the CHU-9D are feasible to use among CYP with CP. However, there is poor agreement between utility values elicited from the two instruments and they should not be used interchangeably to measure and value HRQoL in CYP with CP. This study provides further evidence that it is not appropriate to use the adult EQ-5D tariff to derive utility values from the EQ-5D-Y. Additionally, we propose that the CHU-9D is preferable to the EQ-5D-Y for measuring HRQoL among CYP with CP because it was developed based on interviews with CYP, it assesses concepts that influence QoL among CYP with CP and produces less extreme values than the EQ-5D-Y. However, this study is limited by a small sample size and more research is needed to compare these instruments in CYP with CP and in other clinical populations.

**Contributors** JR and GL conceived the study. JR, GL and NA designed the study. GL, JR, MN and NT acquired data. JR, GL and EM performed the analysis. NA contributed to analysis and interpretation of the data. JR, GL and EM drafted the manuscript. NA, MN and NT critically revised the manuscript. All authors approved the final manuscript.

**Funding** Action Medical Research and the Chartered Society of Physiotherapy Charitable Trust have jointly funded this project, and it is supported by a generous grant from The Henry Smith Charity (GN2340).

**Competing interests** None declared.

**Patient consent for publication** Not required.

**Ethics approval** The study was approved by Brunel University London's College of Health and Life Sciences Research Ethics Committee and the Surrey Borders Research Ethics Committee (ref: 15/LO/0843).

**Provenance and peer review** Not commissioned; externally peer reviewed.

**Data availability statement** Data are available upon reasonable request.

**ORCID iD**
Jennifer M Ryan http://orcid.org/0000-0003-3768-2132

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
