## [Reviewer comments · BMJ Open]

ARTICLE DETAILS

TITLE (PROVISIONAL)	Comparison of the CHU-9D and the EQ-5D-Y instruments in children and young people with cerebral palsy; a cross-sectional study.
AUTHORS	Ryan, Jennifer; McKay, Ellen; Anokye, Nana; Noorkoiv, Marika; Theis, Nicola; Lavelle, Grace

VERSION 1 – REVIEW

REVIEWER	Johan Jarl Lund University, Sweden
REVIEW RETURNED	10-Feb-2020

GENERAL COMMENTS	This study compares the responses to the dimensions of EQ-5D-Y and CHU-9D for an adolescent sample with cerebral palsy. It is found that the instruments are not comparable and CHU-9D is to be preferred in the current population. The authors show that instrument dimensions with similar sounding names in EQ-5D-Y and CHU-9D have in many cases low correlation (i.e. does not seem to capture the same thing). Although this is an important point, I would argue that the main issue is if the utility scores are correlated/similar between instruments. However, this is not studied as there are no child/adolescence tariff developed for EQ-5D-Y. But this begs the question, if the purpose of the current study is to examine which instrument is best to use in an economic evaluation of interventions for adolescents with CP to measure utility/HRQoL, does not already the lack of an appropriate tariff invalidate the EQ-5D-Y? That is, if you cannot calculate a utility score there is no point in using the instrument and CHU-9D is a better choice by default. Alternatively, the question could have been which instrument is preferable when the tariff for EQ-5D is used for EQ-5D-Y (for lack of something better), but then the current presentation should compare the utility scores. When studying the construct validity (that the instruments measure what they are supposed to measure), the authors make the assumption that GMFCS I-III is a good proxy for utility. The appropriateness of this assumption is not discussed. Neither is the fact that no association was found for most dimensions and GMFCS. Does this mean that construct validity of the two instruments is rejected? I would suggest that the authors motivate and discuss this analysis in greater detail (construct validity is currently not mentioned in the discussion section). The authors also set out to compare the performance of the two instruments. This might be a language issue on my part but I
--

	would expect that this type of analysis require something external to which the measurements of the instruments could be compared. The current presentation is more a comparison of the responses than an analysis of the performance of the instruments. With the above in mind, I think that the paper would be much improved by more highlighting the motivation for conducting the current study and outline the expected contribution. Minor comments There is a very large difference between the utility scores obtained in the current study and prior research which is a cause for concern (especially in connection to the GMFCS assumption in the construct validity analysis). Potentially the study sample is highly selected (which could explain some of the unexpected results). Or is there a difference in the tariffs used (hypothetical vs. experience based tariffs where the latter have been shown to give much higher utility values)? Method section mentions histograms, but these are not shown or mentioned in the result section. The text is sometimes a bit wordy, especially when it comes to the comparison between different dimensions. This is easier to see in the table than to follow in the text. Second to last paragraph in result section: the reader needs to be reminded what is being done here and how it was done.
--	---

REVIEWER	Tomos Robinson Newcastle University, United Kingdom
REVIEW RETURNED	17-Feb-2020

GENERAL COMMENTS	Thank you to the authors for submitting this paper. I really enjoyed reading it and definitely think that it has the potential for publication in this journal. However, there are a few changes I would make (mostly minor) before the paper is accepted. I should also note at this point that I am a health economist and don't really know anything about CP. Things to consider:  - I don't really like the inclusion of p-values in the main text, for example line 28 and line 30 of the abstract. This is a personal bugbear, so I understand if other reviewers/the journal think it's worth including them! - In the 'sample' section (line 33), you state several exclusion criteria. I was wondering if you could include a bit more detail about why you chose these criteria? - In the statistical analysis section (line 5) you state that you used linear regression analysis. I was wondering if you could include the full model specification (either in the main text or the appendix) - In the results section (line 30), you state that some participants didn't complete every dimension. I was wondering what you did with these participants? Were they excluded or included in the study?
---

	- In the results section (line 7), you discuss the correlation between 'schoolwork' in the CHU-9D and 'mobility' and 'pain and discomfort'. I'm a bit confused why you did this, as they don't seem to be directly comparable... - Maybe this is me not understanding it properly, but I don't understand how the 'median+-IQR' can be 0.92 +- 0.14, as the CHU-9D utility is bound 0-1? - In the discussion (line 49), I'm a bit confused why you compared the mean values from the Young study to the median values from your study? Why couldn't you compare the mean values? - Finally, I definitely think that you should discuss the fact that you have a small sample size in both the 'Strengths & Limitations' and the 'Discussion' sections. I'm really surprised this wasn't included originally!
--	--

REVIEWER	Anju Devianee Keetharuth School of Health and Related Research University of Sheffield
REVIEW RETURNED	06-Apr-2020

GENERAL COMMENTS	Thanks for the opportunity to review this paper which is a head-to-head comparison of the psychometric properties of CHU-9D and EQ-5D-5L (dimensions) in a cerebral palsy population. The paper is an important addition as it is to authors' knowledge (and mine) the first one to do so. The paper is well written and reads very well. I therefore have only minor comments, several of which are optional.  1. In your sample you have young adults aged 19, therefore I am not sure the term adolescents in the title is strictly correct or is it? I would personally be tempted to use children and young people and then use the acronym (CYP throughout). 2. In the abstract, you may consider including that only dimensions of EQ-5D Y are analysed. This will help in searches and will help researchers reviewing the literature in the future. 3. In the introductory paragraph, you may wish to add the statistics pertaining to children and adolescents which is the focus of this paper. METHODS  4. Can you please be explicit as to whether CHU-9D were self-reported or not in all cases? You mention this was done in an interview but it is not clear to me who completed the measure and how, e.g. were questions read by interviewer etc? 5. Please add cut-offs and reference that you use for assessing the strength of correlation. 6. It will be helpful to set out that you are assessing feasibility (measured by number of missing data <5% with appropriate reference), convergent validity and known-group difference. 7. I am not convinced that "The construct validity of each instrument was examined by comparing the number of people experiencing no problems versus any problems for each CHU-9D and EQ-5D-Y dimension across levels of functional mobility, defined by the GMFCS, using a Chi2 test" is an appropriate way of assessing construct validity. Would a cross-tab not be more meaningful? Known-group difference as assessed by the regression is probably sufficient.
--

	8. “We also examined the association between CHU-9D utility score and levels of each EQ-5D-Y domain by conducting linear regression analysis using a bootstrap procedure.” Can you be more explicit about what your dependent and independent variables? Why did you use the bootstrap procedure? What did it add? RESULTS 9. Typo on Page 7 – line 45 – This should read CHU-9D instead of EQ-5D-Y. 10. Table 4, including n is not very informative. I also wonder whether you can only use asterisks and not include p values. This will make the table more readable to me. It’s just a matter of preference. DISCUSSION 11. It will be helpful to be explicit in the methods section about your hypotheses for the correlation between the two measures and then in the discussion (last para on page 10) refer to whether what you observe are in line with your expectations. 12. “Indeed, this study did not find that functional mobility, as assessed by the GMFCS was associated with overall HRQoL” – would you want to comment a little bit more on that. Is that a problem? Would you have expected it to? If yes, any explanation of why this was not observed. Level 3 GMFCS n = 9 – does this have an implication on your results? 13. One additional limitation is the sample size.
--	---

VERSION 1 – AUTHOR RESPONSE

Reviewer: 1

Reviewer Name: Johan Jarl

Institution and Country: Lund University, Sweden Please state any competing interests or state ‘None declared’: None declared

This study compares the responses to the dimensions of EQ-5D-Y and CHU-9D for an adolescent sample with cerebral palsy. It is found that the instruments are not comparable and CHU-9D is to be preferred in the current population.

Comment:

- The authors show that instrument dimensions with similar sounding names in EQ-5D-Y and CHU-9D have in many cases low correlation (i.e. does not seem to capture the same thing). Although this is an important point, I would argue that the main issue is if the utility scores are correlated/similar between instruments. However, this is not studied as there are no child/adolescence tariff developed for EQ-5D-Y. But this begs the question, if the purpose of the current study is to examine which instrument is best to use in an economic evaluation of interventions for adolescents with CP to measure utility/HRQoL, does not already the lack of an appropriate tariff invalidate the EQ-5D-Y? That is, if you cannot calculate a utility score there is no point in using the instrument and CHU-9D is a better choice by default. Alternatively, the question could have been which instrument is preferable when the tariff for EQ-5D is used for EQ-5D-Y (for lack of something better), but then the current presentation should compare the utility scores.

- When studying the construct validity (that the instruments measure what they are supposed to measure), the authors make the assumption that GMFCS I-III is a good proxy for utility. The appropriateness of this assumption is not discussed. Neither is the fact that no association was found for most dimensions and GMFCS. Does this mean that construct validity of the two instruments is

rejected? I would suggest that the authors motivate and discuss this analysis in greater detail (construct validity is currently not mentioned in the discussion section).

- The authors also set out to compare the performance of the two instruments. This might be a language issue on my part but I would expect that this type of analysis require something external to which the measurements of the instruments could be compared. The current presentation is more a comparison of the responses than an analysis of the performance of the instruments.

- With the above in mind, I think that the paper would be much improved by more highlighting the motivation for conducting the current study and outline the expected contribution.

Response: We aimed to examine the relative performance of the measures. We do not believe this requires comparison to an external measure as we are not aiming to examine criterion validity.

To improve clarity we have stated that “specific objectives were to examine the feasibility of administering to examine convergent validity, and to examine known-group differences for both measures”. We have removed the term construct validity throughout.

We have now calculated utilities for the EQ-5D-Y. In the absence of a preference-based scoring algorithm for the EQ-5D-Y, we used UK-based adult weights to calculate utilities. We examined convergent by calculating an intraclass correlation coefficient and limits of agreement for utilities from the EQ-5D-Y and CHU-9D. We also examined known-group difference by comparing EQ-5D-Y utility values across functional mobility, as indicated by GMFCS level.

We expected that young people with better functional mobility would have higher utilities based on previous research.

Minor comments

Comment: There is a very large difference between the utility scores obtained in the current study and prior research which is a cause for concern (especially in connection to the GMFCS assumption in the construct validity analysis). Potentially the study sample is highly selected (which could explain some of the unexpected results). Or is there a difference in the tariffs used (hypothetical vs. experience based tariffs where the latter have been shown to give much higher utility values)?

Response: As outlined in the discussion, which has been updated, we found a similar median utility value for the EQ-5D-Y as was reported in the two previous studies that compared the EQ-5D-Y and CHU-9D in children and adolescents with typical development. We also observed a similar median utility value for the CHU-9D as was reported for children with typical development living in England. The two previous studies that reported utility values in children and young people with CP used the HUI-3 and not the EQ-5D-Y and CHU-9D. The mean utility values reported for each GMFCS level differed between these two studies. The mean EQ-5D-Y utility values we calculated for each GMFCS level were similar to those reported by Rosenbaum. We believe the higher mean CHU-9D utility value that we calculated for each GMFCS level may be more accurate than the mean values calculated from the EQ-5D-Y and HUI-3 as the CHU-9D was developed specifically for children and young people.

Comment: Method section mentions histograms, but these are not shown or mentioned in the result section.

Response: We originally plotted histograms to assess the normality of data, which informed which descriptive statistics were reported. However, we have now added the histograms to the results.

Comment: The text is sometimes a bit wordy, especially when it comes to the comparison between different dimensions. This is easier to see in the table than to follow in the text.

Response: We have reduced the text in relation to comparison between different dimensions and refer the reader to Table 4.

Comment: Second to last paragraph in result section: the reader needs to be reminded what is being done here and how it was done.

Response: We have rewritten the whole results section to improve clarity.

Reviewer: 2

Reviewer Name: Tomos Robinson

Institution and Country: Newcastle University, United Kingdom Please state any competing interests or state 'None declared': Non Declared

Please leave your comments for the authors below Thank you to the authors for submitting this paper. I really enjoyed reading it and definitely think that it has the potential for publication in this journal. However, there are a few changes I would make (mostly minor) before the paper is accepted. I should also note at this point that I am a health economist and don't really know anything about CP.

Comment: I don't really like the inclusion of p-values in the main text, for example line 28 and line 30 of the abstract. This is a personal bugbear, so I understand if other reviewers/the journal think it's worth including them!

Response: We have removed the p values from the main text (abstract and results sections).

Comment: In the 'sample' section (line 33), you state several exclusion criteria. I was wondering if you could include a bit more detail about why you chose these criteria?

Response: This study was nested within a randomised controlled trial (RCT) examining the feasibility, acceptability and efficacy of resistance training for adolescents with cerebral palsy. The data for this study was collected as part of the routine assessments of this larger RCT and therefore the specified exclusion criteria related to the original protocol. The recruitment and selection criteria have been previously reported in Ryan et al. (10.1136/bmjopen-2016-012839). We have added that the inclusion and exclusion criteria were for participation in the trial and referenced the trial protocol.

Comment: In the statistical analysis section (line 5) you state that you used linear regression analysis. I was wondering if you could include the full model specification (either in the main text or the appendix)

Response: We have put the output from the models examining the association between utilities and GMFCS level in supplemental tables 1 and 2. We have removed the linear regression examining associations between CHU-9D utility values and levels on each EQ-5D-Y dimension. We now report Spearman's correlation coefficients to improve ease of interpretation.

Comment: In the results section (line 30), you state that some participants didn't complete every dimension. I was wondering what you did with these participants? Were they excluded or included in the study?

Response: Sixty-four participants were recruited to the study. One person did not complete the EQ-5D-Y or CHU-9D. Therefore, 63 participants were included in the analysis. All 63 participants included in the final analysis in this paper completed both the EQ-5D-Y and CHU-9D questionnaires. However, some participants failed to answer each individual dimension within each of the questionnaires. All available data provided was used. Data were only compared and reported on between measures where data on both scales was available. For the CHU-9D, two participants (3.2%) did not provide a response to the schoolwork dimension and one participant (1.6%) did not provide a response to the ability to "join in activities" dimension. For the EQ-5D-Y, one participant (1.6%) did not provide a response to the "looking after myself" dimension. Therefore, the sample sizes for analysis involving

these items reduced accordingly. These individuals were also excluded from calculation of utility scores. Table 4 indicates the sample sizes included in each analysis. We have added the following sentence to the statistical analysis section: Participants who were missing data for an individual dimension of the EQ-5D-Y or CHU-9D were excluded from the calculation of utility scores and from analyses of that dimension.

Comment: In the results section (line 7), you discuss the correlation between 'schoolwork' in the CHU-9D and 'mobility' and 'pain and discomfort'. I'm a bit confused why you did this, as they don't seem to be directly comparable...

Response: We examined associations between all dimensions of the EQ-5D-Y and CHU-9D. We have added that we hypothesised certain dimensions would be related (as outlined in the statistical analysis section). However, we observed some associations that we did not expect. These include schoolwork vs mobility and schoolwork vs pain/discomfort. We have discussed potential reasons for these associations in the discussion.

Comment: Maybe this is me not understanding it properly, but I don't understand how the 'median+-IQR' can be 0.92 +- 0.14, as the CHU-9D utility is bound 0-1?

Response: The 25th percentile is 0.83 and the 75th percentile is 0.96. The interquartile range is the difference between these (i.e., 0.13). To make it clearer to interpret we have now presented the IQR as the 25th percentile to 75th percentile (i.e., 0.83 to 0.96).

Comment: In the discussion (line 49), I'm a bit confused why you compared the mean values from the Young study to the median values from your study? Why couldn't you compare the mean values?

Response: We did not calculate the mean value in our study as data were not normally distributed. However, we now present the mean and median utilities to allow comparison with mean values from Young.

Comment: Finally, I definitely think that you should discuss the fact that you have a small sample size in both the 'Strengths & Limitations' and the 'Discussion' sections. I'm really surprised this wasn't included originally!

Response: We have added the small sample as a limitation to the 'Strengths & Limitations' and the 'Discussion' sections.

Reviewer: 3

Reviewer Name: Anju Devianee Keetharuth

Institution and Country:

School of Health and Related Research

University of Sheffield

Please state any competing interests or state 'None declared': None declared

Thanks for the opportunity to review this paper which is a head-to-head comparison of the psychometric properties of CHU-9D and EQ-5D-5L (dimensions) in a cerebral palsy population. The paper is an important addition as it is to authors' knowledge (and mine) the first one to do so. The paper is well written and reads very well. I therefore have only minor comments, several of which are optional.

Comment: In your sample you have young adults aged 19, therefore I am not sure the term adolescents in the title is strictly correct or is it? I would personally be tempted to use children and young people and then use the acronym (CYP throughout).

Response: We agree with this suggestion and have changed adolescents to children and young people in the title and throughout the main body of the text also, where referring to participants in this study.

Comment: In the abstract, you may consider including that only dimensions of EQ-5D Y are analysed. This will help in searches and will help researchers reviewing the literature in the future.

Response: We have now analysed EQ-5D-Y utility values and have updated the abstract accordingly.

Comment: In the introductory paragraph, you may wish to add the statistics pertaining to children and adolescents which is the focus of this paper.

Response: In the introductory paragraph we present statistics on the number of people with CP in the UK (approximately 110,000), according to the National Service Framework for Long-Term Conditions (2005). Unfortunately, this report did not provide a breakdown of number per age and to the authors' knowledge this information not been specifically reported upon here or elsewhere to include.

METHODS

Comment: Can you please be explicit as to whether CHU-9D were self-reported or not in all cases? You mention this was done in an interview but it is not clear to me who completed the measure and how, e.g. were questions read by interviewer etc?

Response: We have added the following text to clarify the mode of administration:

Both HRQoL questionnaires were self-administered to all participants using the standardised instructions accompanying each measure. Assistance was provided by the researcher to read the questions if required. Further, the young person was allowed to ask their parent/guardian or researcher for assistance to answer the questions if required.

Comment: Please add cut-offs and reference that you use for assessing the strength of correlation.

Response: We have added these with references to the statistical analysis section and stated that this interpretation of the strength of associations should be used with caution.

Comment: It will be helpful to set out that you are assessing feasibility (measured by number of missing data <5% with appropriate reference), convergent validity and known-group difference.

Response: We have now stated that specific objectives are to examine feasibility, convergent validity and known-group difference. We have also updated the statistical analysis section to indicate how we assessed each of these.

Comment: I am not convinced that "The construct validity of each instrument was examined by comparing the number of people experiencing no problems versus any problems for each CHU-9D and EQ-5D-Y dimension across levels of functional mobility, defined by the GMFCS, using a Chi2 test" is an appropriate way of assessing construct validity. Would a cross-tab not be more meaningful? Known-group difference as assessed by the regression is probably sufficient.

Response: We have removed the term construct validity throughout and stated that we are assessing feasibility, convergent validity and known-group differences. We have kept the comparison of those experiencing no problems versus any problems for each dimension across GMFCS level as we believe this is of interest to readers. We did not conduct a Chi2 test to examine the association between each level of each dimension and GMFCS level because a number of levels had 0 counts. However, we present the percentage of people in each level for each dimension across GMFCS levels I-III in tables 5 and 6.

Comment: "We also examined the association between CHU-9D utility score and levels of each EQ-5D-Y domain by conducting linear regression analysis using a bootstrap

procedure.” Can you be more explicit about what your dependent and independent variables? Why did you use the bootstrap procedure? What did it add?

Response: We have removed the linear regression examining associations between CHU-9D utility values and levels on each EQ-5D-Y dimension. We now report Spearman’s correlation coefficients to improve ease of interpretation.

For the linear regression examining associations between utilities and GMFCS level we have stated that utility value was the dependent variable and that we used a bootstrap procedure because there was evidence that residuals were not normally distributed. We have also added a table with the output from each model examining the association between utilities and GMFCS level to the appendix.

RESULTS

Comment: Typo on Page 7 – line 45 – This should read CHU-9D instead of EQ-5D-Y.

Response: We have checked the text and EQ-5D-Y is the correct measure being referenced in this line.

Comment: Table 4, including n is not very informative. I also wonder whether you can only use asterisks and not include p values. This will make the table more readable to me. It’s just a matter of preference.

Response: We have removed n from the table and indicated n with a footnote where $n < 63$. We would prefer to keep the p values in the table to indicate the strength of the evidence against the null hypothesis.

DISCUSSION

Comment: It will be helpful to be explicit in the methods section about your hypotheses for the correlation between the two measures and then in the discussion (last para on page 10) refer to whether what you observe are in line with your expectations.

“Indeed, this study did not find that functional mobility, as assessed by the GMFCS was associated with overall HRQoL” – would you want to comment a little bit more on that. Is that a problem? Would you have expected it to? If yes, any explanation of why this was not observed. Level 3 GMFCS $n = 9$ – does this have an implication on your results?

Response: We have updated the discussion in line with the revised objectives, statistical analysis and results. We have discussed the results in relation to our expected findings and have commented on the small number of people in GMFCS level III.

Comment: One additional limitation is the sample size.

Response: We have added this to the discussion and strengths and limitations sections.

VERSION 2 – REVIEW

REVIEWER	Johan Jarl Lund University, Sweden
REVIEW RETURNED	25-May-2020

GENERAL COMMENTS	The authors have successfully revised the manuscript. On a very minor note, I would suggest the following to be revised: * Page 7, line 34: explain what the Bland-Altman plot is/does * Page 8, lines 54+: help the reader interpret the Bland-Altman plot
---

	* Page 9, lines 3-10: I think the tables have been switched (table 2 should be table 3 and vice versa) * Page 9, line 22: typo * Page 9, line 37: typo * Page 11, line 5: repetition from two lines up
--	--

REVIEWER	Tomos Robinson Newcastle University
REVIEW RETURNED	14-May-2020

GENERAL COMMENTS	Thank you to the authors for updating the manuscript. You have integrated all my suggested changes into the manuscript very well. From looking through the other reviewer's suggestions, it seems as though you have integrated their comments well too. I do have one more (very small) suggestion. I know I discussed the small sample size in the previous round of corrections, but I'm not entirely comfortable with you stating "We propose that the CHU-9D is preferable to the EQ-5D-Y for measuring HRQoL among CYP with CP..." in the discussion without saying something like "however more work is needed in this area", given your small sample size. Aside from this, I think the paper is ready for publication and I don't need to see it again!
---

REVIEWER	Anju Keetharuth University of Sheffield
REVIEW RETURNED	28-May-2020

GENERAL COMMENTS	Thanks a lot for addressing all the comments in a satisfactory manner. Now that you have calculated utilities for EQ-5D-Y using the value set, you need to acknowledge the limitation with doing this. These two references deal with this issue: Kind P, Klose K, Gusi N, Olivares PR, Greiner W. Can adult weights be used to value child health states? Testing the influence of perspective in valuing EQ-5D-Y. Quality of Life Research. 2015 Oct 1;24(10):2519-39. Kreimeier S, Oppe M, Ramos-Goñi JM, Cole A, Devlin N, Herdman M, Mulhern B, Shah KK, Stolk E, Rivero-Arias O, Greiner W. Valuation of EuroQol five-dimensional questionnaire, youth version (EQ-5D-Y) and EuroQol five-dimensional questionnaire, three-level version (EQ-5D-3L) health states: the impact of wording and perspective. Value in Health. 2018 Nov 1;21(11):1291-8.w
---

VERSION 2 – AUTHOR RESPONSE

Reviewer: 2

Thank you to the authors for updating the manuscript. You have integrated all my suggested changes into the manuscript very well. From looking through the other reviewer's suggestions, it seems as though you have integrated their comments well too.

I do have one more (very small) suggestion. I know I discussed the small sample size in the previous round of corrections, but I'm not entirely comfortable with you stating "We propose that the CHU-9D is preferable to the EQ-5D-Y for measuring HRQoL among CYP with CP..." in the discussion without saying something like "however more work is needed in this area", given your small sample size.

Response: Thank you for reviewing this manuscript again and for your positive feedback. We have added the following sentence to the conclusion "However, this study is limited by a small sample size and more research is needed to compare these instruments in CYP with CP and in other clinical populations."

Reviewer: 1

The authors have successfully revised the manuscript. On a very minor note, I would suggest the following to be revised:

* Page 7, line 34: explain what the Bland-Altman plot is/does

Response: We have added the following "We also produced a Bland-Altman plot of the difference between the two instruments against their mean to examine agreement between utilities from the two instruments. We calculated 95% limits of agreement as mean difference $\pm 1.96SD$ ". We also included a reference to Bland JM, Altman DG. Statistical methods for assessing agreement between two methods of clinical measurement. Lancet 1986;327:307-10.

* Page 8, lines 54+: help the reader interpret the Bland-Altman plot

Response: We have added the following: 95% limits of agreement were -0.58 to 0.29, indicating that the EQ-5D-Y utility value may be 0.29 higher or 0.58 lower than the CHU-9D utility value (Figure 3).

* Page 9, lines 3-10: I think the tables have been switched (table 2 should be table 3 and vice versa)

Response: Thank you for spotting this error. We have reordered tables 2 and 3.

* Page 9, line 22: typo

* Page 9, line 37: typo

Response: Thank you for spotting these typos. We have corrected them.

* Page 11, line 5: repetition from two lines up

Response: Thank you for noticing this. We have removed this sentence.

Reviewer: 3

Thanks a lot for addressing all the comments in a satisfactory manner.

Now that you have calculated utilities for EQ-5D-Y using the value set, you need to acknowledge the limitation with doing this. These two references deal with this issue:

Kind P, Klose K, Gusi N, Olivares PR, Greiner W. Can adult weights be used to value child health states? Testing the influence of perspective in valuing EQ-5D-Y. *Quality of Life Research*. 2015 Oct 1;24(10):2519-39.

Kreimeier S, Oppe M, Ramos-Goñi JM, Cole A, Devlin N, Herdman M, Mulhern B, Shah KK, Stolk E, Rivero-Arias O, Greiner W. Valuation of EuroQol five-dimensional questionnaire, youth version (EQ-5D-Y) and EuroQol five-dimensional questionnaire, three-level version (EQ-5D-3L) health states: the impact of wording and perspective. *Value in Health*. 2018 Nov 1;21(11):1291-8.

Response: Thank you for providing us with these references. We have included the following sentence in the limitations: "The use of adult weights to value EQ-5D-Y health states is a limitation of this study. It has previously been demonstrated that values for health states ascribed by adults differ to corresponding health states ascribed by children [32,33]. However, we used adult weights in the absence of value sets for the EQ-5D-Y."